

# Determining large hyperfine interactions of a model flavoprotein in the semiquinone state by pulse-EPR techniques

Jesús I. Martínez[1,3], Susana Frago[2], Milagros Medina[2], Inés García-Rubio[3,4]

[1]Departmento de Física de la Materia Condensada, Universidad de Zaragoza, Zaragoza, 50009, Spain
[2]Departmento de Bioquímica y Biología Molecular y Celular and Instituto de Biocomputación y Física de Sistemas Complejos (BIFI), Universidad de Zaragoza, Zaragoza, 50009, Spain
[3]Instituto de Ciencia de Materiales de Aragón, CSIC-Universidad de Zaragoza, 50009, Spain
[4]Institute for Molecular Physical Science, ETH Zurich, 8093 Zürich, Switzerland

*Correspondence to*: Inés García-Rubio (inesgr@unizar.es)

**Abstract.**

Flavoproteins are a versatile class of proteins involved in numerous biological processes, including redox reactions, electron transfer, and signal transduction, often relying on their ability to stabilize different oxidation states of their flavin cofactor. A critical feature of flavin cofactors is their capacity to achieve, within particular protein environments, a semiquinone state

that plays a pivotal role in mediating single-electron transfer events and is key to understanding flavoprotein reactivity.

Hyperfine interactions between the unpaired electron in the semiquinone state and magnetic nuclei in the isoalloxazine ring provide valuable insights into the electronic structure of this intermediate and its mechanistic roles. This study investigates the hyperfine interactions of isotopically labeled flavodoxin (Fld) with $^{13}$C and $^{15}$N at specific positions of the flavin mononucleotide (FMN) ring using advanced electron paramagnetic resonance (EPR) techniques. The combination of

Continuous wave (CW) EPR at X-band and ELDOR-detected NMR and HYSCORE at Q-band revealed a strong and anisotropic hyperfine interaction with the nucleus $^{13}$C at 4a and yielded principal tensor values of 40, -13.5 and -9 MHz, the first of which is associated to the axis perpendicular to the flavin plane. On the other hand, as predicted, the hyperfine interaction with the $^{13}$C nucleus at position 2 was minimal. Additionally, HYSCORE experiments on $^{15}$N-FMN labeled Fld provided precisely axial hyperfine parameters, (74, 5.6, 5.6) MHz for $^{15}$N(5) and (38, 3.2, 3.2) MHz for $^{15}$N(10). These were

used to refine quadrupole tensor values for $^{14}$N nuclei via isotope-dependent scaling. These results showcase the potential of combining CW-EPR, ELDOR-detected NMR, and HYSCORE with isotopic labelling to probe electronic and nuclear interactions in flavoproteins. The new data complete and refine the existing experimental map for the electronic structure of the flavin cofactor and expose systematic divergences between the calculated and experimental values of hyperfine couplings of the atoms most contributing to the SOMO. This could indicate a slight but significant shift of the unpaired electron

density from position 4a towards the central nitrogens of the pyrazine ring as compared to the calculations. These results highlight the importance of integrating computational and experimental approaches to refine our understanding of flavin cofactors reactivity.



## 1 Introduction

Flavoproteins constitute an extensive and versatile family of proteins which are involved in a wide range of biochemical and physiological processes. These include, among others, the catalysis of different redox reactions, the support of electron transfer chains, the ability to act as sensors of the redox state of the cell, oxygen or voltage, and the specific capability to activate catalytic processes upon their activation by blue light or magnetoreception (Walsh and Wencewicz, 2013, Minjarez-Saez et al., 2022, Calloni and Vabulas, 2023, Moreno, A. et al., 2024, Kaya et al., 2015, Matysik et al., 2023, Fraaije and

Mattevi, 2000).

Flavoproteins have as cofactor a group that includes the 7,8-dimethyl-isoalloxazine ring, usually in the form of Flavin mononucleotide (FMN) or Flavin adenine dinucleotide (FAD). This ring is able to reach three different oxidation states: fully oxidized, fully reduced by two electrons, and an intermediate semiquinone state holding an additional electron over the oxidized state and, therefore, having an unpaired electron and being paramagnetic. The semiquinone state is usually not

detectable in redox processes involving free flavins, but in some flavoproteins it is highly stabilized by the protein environment. This is an indication that the combination of the unique properties of ixoalloxazine ring and of the modulation of its redox and electronic properties by the protein environment configures its ability to develop different functions in each specific flavoprotein. In particular, the ability of some flavoproteins to stabilize the semiquinone state during their redox cycle makes them unique molecules to mediate processes involving donor/acceptors with the ability of exchange only one

electron at a time (in general metal centers), with those that obligatorily exchange two electrons (as for example pyridine nucleotides) (Lans et al., 2012, Young et al, 2020; Moreno et al., 2024).

The spin density distribution in the semiquinone state of flavin cofactors has a clear relation with protein reactivity and its characterization can provide insights about electron tranfer pathways. A better knowledge of the orbital composition of the SOMO of the semiquinone state is of great mechanistic relevance, because the reactivity of the fully reduced state to

generate it and the topology of the electron transfer process depend entirely on it. Also the life time of this semiquinone intermediate and the way it transits to the completely oxidized state for processes occurring in the different flavoproteins are intimately related to such electronic structure. Hyperfine interactions between the unpaired electron and the magnetic nuclei in the cofactor are directly related to the orbital composition of the semioccupied orbital (SOMO) and can be used to probe electron density. Moreover, hyperfine interactions are also relevant for the elucidation of the flavoproteins mechanisms

themselves as they influence, for instance, flavin magnetochemistry in magnetorecepcion or the lifetime of semiquinone intermediates. Therefore, numerous reports have aimed at the characterization of hyperfine structure using different techniques of Electron Paramagnetic Resonance on several flavoproteins (Weber et al., 2005; Schleicher et al., 2009; Martínez et al., 2012; Schleicher et al., 2012; Brosi et al., 2014; Nohr et al., 2019; Weber et al., 2021).



The effect of hyperfine interactions can be observed already in the CW-EPR spectra of the flavin, however, characterizations
by hyperfine spectroscopy methods able to provide higher resolution (ESEEM-HYSCORE or ENDOR) have revealed more
details about the anisotropy of the interaction. The use of different techniques and microwave frequencies, together with
model flavoproteins containing isotopically enriched flavins in selected positions or flavin analogues, has allowed to obtain
the complete hyperfine tensors for many nuclei at the isoalloxazine ring. Most reported hyperfine interactions refer to
protons within the ring. Although they provide relevant information, these interactions are just indirectly related to spin
densities at the sites of the ring to which they are attached and, therefore, are less significant compared to the interactions
with nuclei directly part of the ring. To probe the electronic distribution on the carbon atoms on the isoalloxazine, the use of
flavin samples with flavins isotopically labelled is necessary, since the most abundant carbon nucleus $^{12}$C has no nuclear
spin, $I_{12C} = 0$. For nitrogen positions, over 99% of nuclei are $^{14}$N, with a nuclear spin $I_{14N} = 1$ that usually displays
experimental evidence that is difficult to interpret. Therefore, labelling with $^{15}$N ($I_{15N} = 1$) is also helpful. As a result of the
combination of high-resolution methods and isotopic labelling, information on couplings to nitrogen nuclei in the ring has
been obtained through HYSCORE experiments at X band, CW-EPR at W band and pulse ENDOR at W band (Martínez et
al., 1997; Barquera et al., 2003; Weber et al., 2005). Additionally, a detailed study has also been published on a protein with
a flavin cofactor selectively enriched with $^{13}$C at different carbon sites, using the ENDOR pulse technique at W band
(Schleicher et al., 2021). However, the characterization of the hyperfine interaction with some of the nuclei at the sites with
the highest spin densities, namely positions C(4a) and N(5) (see Fig. 1 for site numbering), still remains incomplete,
preventing the experimental determination of the spin population. This information is also of particular interest from the
flavoproteins and flavoenzymes functional point of view, because the N(5)-C(4a) locus of the flavin concentrates most of its
chemical prowess. In fact, the N(5) of the isoalloxazine is known to be relevant during redox processes and substrate
oxygenation, while C(4a) is a recognized site for one-electron chemistry during flavin reoxidation processes (Beaupre &
Moran, 2021, Sucharitakul et al., 2011, Ghisla & Massey, 1989, Visitsatthawong et al., 2015, Saleem-Batcha, 2018).
Furthermore, the reactivity of N(5) and C(4a) allows formation of covalent intermediates contributing to increase the
chemical repertoire of natural flavin derivatives within flavoproteins (Leys and Scrutton, 2016, Beaupre and Moran, 2020).

In parallel to these studies, calculations, mainly based on DFT computational methods, have been published in the last
twenty-five years to improve the knowledge of the electronic structure of the flavin cofactor in its three oxidation states
(HOMO and LUMO of the completely oxidized and reduced states, and SOMO of the semiquinone state) aiming to predict
their physicochemical properties (Domratcheva et al., 2014). Although these methods have a great capacity to develop
realistic electronic structures, considering in detail the effect of the environment, obviously their validity must be contrasted
with experimental results and the discrepancy must be used to improve or refine the results of those calculations. Hyperfine
interactions of the unpaired electron in the semiquinone state with nuclei in the vicinity of the isoalloxazine ring are very
valuable experimental control parameters to contrast the result of the calculations, as these interactions are directly related to
the spin density distribution in the SOMO. They provide an appropriate tool to test the predictions of the calculations for the



electronic structure of the semiquinone state, as the magnetic nuclei act as local probes for drawing a map of the electronic spin density within the ring.

Over these years it has been recognized though the calculation of different flavoproteins (such as Fld and DNA photolyase) that the values for the hyperfine parameters obtained through calculations are very sensitive to the level of calculation and the functional used, as well as to the environment of the flavin that is considered (Weber et al., 2002, García et al., 2002, Schleicher et al., 2021). However, experimental evidences indicate that hyperfine interactions are very similar in different flavoproteins (Barquera et al., 2003; Martínez et al., 2012,; Schleicher et al., 2012; Paulus, 2014; Nohr et al., 2019), and that the substitution of residues in the flavin environment or in the ring atoms barely affects the hyperfine splittings (Medina et al., 1999, Martínez et al., 2016). In general, the experimental differences between different flavoproteins are in the same order or smaller than those that appear between calculations depending on their technical details (level, functional used and environment considered). The agreement between the calculations and the known experimental values of $a_{\mathrm{iso}}$ is fair for interactions with [1]H nuclei (most of the calculated values are within ± 20% of the experimental values) but the hyperfine couplings with [14]N(5) and [14]N(10) nuclei appear to be systematically underestimated, whereas the relative error remains within ± 20% for the completely characterized [13]C nuclei within the flavin ring[1] (Schleicher at al, 2021).

In the analysis of the hyperfine interactions describing the electronic structure of the flavin semiquinone radical for a better understanding of its role in flavoprotein-catalyzed reactions, it is certainly worth including the hyperfine interactions with nitrogen nuclei. Nitrogen occupies four positions within the ring, namely N(1), N(3), N(5) and N(10), for which previously reported experimental evidence of the hyperfine couplings is available (Martínez et al., 97; Barquera et al., 2003; Weber et al., 2005; Martínez et al., 2012). Furthermore, the anisotropic part of the hyperfine also provides relevant information that should not be neglected. In a pure $\pi$ radical, the hyperfine interaction with nuclei on the ring is axial, being the axis perpendicular to the plane of the ring. The detection of orthorhombic hyperfine matrices or hyperfine principal values that show different proportions between the isotropic and anisotropic part implies the mixture of $\pi$ and $\sigma$ orbitals linked to distortion of the molecular and/or electronic structure that can have a relevant effect on the mechanisms where flavin is involved. Besides, although this evidence does not directly inform the electronic structure of the fully reduced and oxidized states, the disparity between the calculated and measured hyperfine splitting values offers an indirect indication about differences that may exist between the electronic structures calculated for those states and real ones.

---

[1] The relative error for some weakly coupled nuclei is larger than this value. However, since the absolute values of the discrepancies are small they are considered not very relevant.





**Figure 1: Sketch of the molecular structure** and IUPAC numbering of a (7,8-dimethyl) isoalloxazine neutral semiquinone radical.

In this work we present an X and Q band study of the neutral semiquinone of *Anabaena* Fld combining CW-EPR, ELDOR detected NMR and HYSCORE experiments with selective $^{13}$C and $^{15}$N isotope labelling of the flavin. Fld was chosen as a model system due to its feasibility to replacing its FMN cofactor with modified flavins and to its ability to stabilize a large

proportion of its neutral semiquinone state (Martínez et al., 2012; Martínez et al., 2014; Martínez et al., 2016, Lans et al., 2012). This particular combination of experiments and microwave frequencies (about 9 and 34 GHz) has turned out to be especially suited for the detection of couplings at the C(4a), N(5) and N(10) positions of the isoalloxazine, allowing the complete characterization of the hyperfine interactions for the nuclei $^{13}$C(4a), $^{15}$N(5) and $^{15}$N(10). These results provide with a suitable protocol to experimentally access these couplings and an estimation of the spin density in the Fld model. These

results are discussed based on the predictions of published calculations and on our knowledge of electron transfer processes involving flavoproteins.

## 2 Materials and Methods

### 2.1 Biological Material

Riboflavin (RF) analogues $^{13}$C(2)-RF and $^{13}$C(2, 4a)-RF  were converted into the corresponding FMN forms using the mutant H28A of FAD synthase (FADS) from *Corynebacterium ammoniagenes* (Frago et al., 2010, Frago et al., 2008), Reaction mixtures containing 50 µM of the RF analogue, 0.5 mM ATP, 1 mM MgCl$_2$ and 1.5–3 µM H28A FADS in 50 mM Tris/HCl, pH 8.0, were incubated in the dark at 37 °C overnight. Full conversion of RF into FMN was checked by thin layer chromatography in silica-gel plates. Once the reaction was completed, FADS was separated from the flavin by ultrafiltration

(Amicon Ultra, Millipore, 10000 MW cut-off). $^{15}$N-labeled FMN was produced as previously described (Martinez et al., 1997), *Anabaena* Fld was over-expressed in *Escherichia coli* and purified as described in Genzor at al., 1996. ApoFld was prepared by treatment with 3% trichloroacetic acid at 4 °C in the presence of dithiothreitol. The precipitated apoprotein was separated from FMN by centrifugation and dissolved in 500 mM MOPS pH 7.0 before dialysis against 50 mM MOPS pH 7.0. Finally, *Anabaena* ApoFld was incubated with a 1.5-fold molar excess of each FMN analogue (namely $^{13}$C(2)-FMN,



$^{13}$C(2, 4a)-FMN or $^{15}$N-FMN) in 50 mM MOPS, pH 7.0 for 1 h at 25 °C. Excess flavin was then removed by ultrafiltration and the reconstituted Flds stored at −20 °C. Samples with a protein concentration of 400–800 mM in 50 mM MOPS, pH 7.0, were placed in 3 mm EPR tubes and anaerobically reduced under an argon atmosphere to the semiquinone state at 4 °C by light irradiation with a 150 W Barr & Stroud light source, approximately 7.5 cm from the sample, in the presence of 20 mM EDTA and 2.5 $\mu$M 5-deazariboflavin. Once maximal production of the neutral semiquinone was obtained, samples were

frozen and stored in liquid nitrogen (at 77 K) until used in EPR measurements.

**2.2 EPR spectroscopy**

X-band EPR experiments were performed in a Bruker Elexys E580 spectrometer (microwave frequency ∼ 9.7 GHz) equipped with a cylindrical dielectric cavity and a helium gas-flow cryostat from Oxford Inc. Q-band pulse EPR measurements were carried out on a home-built spectrometer operational in the frequency range of 34.5−35.5 GHz (Gromov

et al., 2001) equipped with a custom-made resonator allowing the use of 3 mm sample tubes (Tschaggelar et al., 2009). The spectra were taken at 50 or 90 K. The repetition rate was generally 3 kHz. The HYSCORE and ELDOR-detected NMR experiments were carried out at different observer positions that correspond to different selections of orientations of the molecules with respect to the magnetic field.

2.2.1 CW-EPR

The X-band CW-EPR spectra were acquired at 9.714GHz at a temperature of 50 K, using a modulation amplitude of 2 G and a microwave power of 0.32 μW. The Q-band Electron Spin Echo (ESE)-detected EPR spectra were detected using a $\pi/2−\tau−\pi−\tau−echo$ sequence with pulse lengths of 16 and 32 ns for the $\pi/2$ and $\pi$ pulses. The Q-band Free Induction Decay (FID)-detected EPR spectra were recorded with the pulse sequence $\pi$ –FID, where the mw pulse was 1μs and the FID was integrated over its whole duration.

2.2.2 HYSCORE

*HYSCORE* experiments (Höfer, 1994, Schweiger et Jeschke, 2001) were performed at Q-band frequency (34.3 GHz) at a temperature of 50 K ($^{13}$C-labelled samples) or 90 K ($^{15}$N-labelled samples) using the pulse sequence $\pi/2−\tau−\pi/2−t_1−\pi−t_2−\pi/2−\tau−echo$. Different $\tau$ values were used and specified in the figure captions. Unless stated otherwise, pulse lengths of 12 ns for all pulses was used in the experiments performed at the echo maximum to obtain maximum

excitation width and 24 and 16 ns pulse lengths were programmed for $\pi/2$ and $\pi$ pulses respectively for the experiments performed at the high-field or low-field flanks of the CW spectrum to obtain better orientation selectivity. The time intervals $t_1$ and $t_2$ were varied in steps of 8, 12 or 16 ns starting from 96 ns. A phase cycle of eight steps was used to eliminate unwanted echoes. The experimental time traces were baseline corrected, apodized with a Hamming or Gaussian window,





and zero filled. After a Fourier transformation in the two time dimensions, the absolute-value spectra were calculated and
plotted with Matlab.

### 2.2.3 ELDOR-detected NMR

The Q-band *ELDOR-detected NMR* (Schosseler et al., 1994, Schweiger and Jeschke, 2001) experiments were performed
using the pulse sequence $(HTA)_{mw2}−τ−(π)_{mw1}−FID$. Two rectangular pulses were used. The pulse lengths were 1 μs for the
first pulse with variable mw frequency $(mw_2)$, and 1 μs for the second pulse with fixed mw frequency $mw_1$. The separation
between the two pulses was $τ = 1.5$ μs. The FID generated after the second pulse was integrated over a width of 800 ns. The
real and imaginary parts were acquired, baseline shifted, and the absolute value was calculated. The spectra were inverted
(multiplied by -1) for display.

### 2.3 Spectral simulations

CW-EPR and HYSCORE spectra were simulated using the toolbox for MATLAB EasySpin (Stoll and Schweiger, 2006),
version 6.0.6 freely downloadable from www.easyspin.org, using the functions *pepper* and *saffron* respectively and the spin
Hamiltonian specified below. For the HYSCORE simulations, in a first step the nuclear frequencies were computed using
the function *endorfreq* and the Hamiltonian given in eq 1, whereby the orientation selection of the experiment was taken into
account. From these frequencies, the position and shape of the HYSCORE correlation ridges of the individual nuclei can be
deduced, but no information is obtained about the intensity of the cross-peaks. In a second step, when all the transitions were
identified HYSCORE simulations of the complete nuclear system were done using the function saffron of EasySpin
providing the combination ridges and the intensities of all the lines. The time-domain simulations were processed and plotted
using the same procedure as the experimental data described above.

### 2.4 Spin Hamiltonian

 The Spin Hamiltonian (SH) that was used to analyze the experimental spectra and characterize the hyperfine interactions of
the semiquinone radical $(S = 1/2)$ with $n$ different nuclear spins $(I_i)$ in the isoaloxazine ring consists of several terms:

$$H = \mu_B \vec{B} \hat{g} \vec{S} + \sum_i \mu_n \vec{B} g_n \vec{I_i} + \vec{S} \hat{A_i} \vec{I_i} + \sum_{I_i > 1/2} \vec{I_i} \hat{Q_i} \vec{I_i} \qquad (1)$$

The first and second terms of the SH represent the electron and nuclear Zeeman interactions, respectively. In the case of
semiquinome radicals, the electron g-tensor is close to the free electron g-factor, as expected for a radical. However, at
higher mw frequencies some anisotropy in the tensor can be resolved (Fuchs et al., 2002; Kay et al., 2005; Okafuji, 2008). In
this article we assumed $(g_z = 2.0022, g_y = 2.0036$ and $g_x = 2.0043)$. The third term takes into account the hyperfine



interactions with the different magnetic nuclei. If $I > 1/2$, as is the case for $^{14}$N ($I = 1$), the nuclear-quadrupole interaction has to be included (fourth term). $A_i$ and $Q_i$ are the hyperfine and nuclear quadrupole tensors, respectively, of nucleus $i$.

Reflecting the planar symmetry of the molecule, the hyperfine tensors of magnetic nuclei directly in the flavin ring have been reported to be mostly axial, with the separate axis perpendicular to the ring (normally called $z$). Since most of the electron density is located in $\pi$ orbitals, the hyperfine interaction along this direction use to be sensibly larger than in the other two directions contained in the plane. For moderate mw frequencies (X Band and below) this results in a CW-EPR spectrum with a central intense line corresponding to the perpendicular features and low- and high-field wings that are

contributed mainly by the molecules oriented with their axis parallel to the magnetic field. On these wings, some ripples due to large (parallel) hyperfine couplings can be resolved (Martínez et al., 2016). The distance between the two outermost features (highest-field, labelled O2 in Fig. 2, and lowest-field, labelled O1 in Fig. 2) is the sum of the couplings of all magnetic nuclei in the direction perpendicular to the isoaloxacine plane. For flavins with natural isotopic abundance, the three nuclei with the largest couplings in semiquinone radicals are $^{14}$N(5), $^{1}$H(5) and $^{14}$N(10), therefore, neglecting smaller

couplings, the distance between the two outermost shoulders is approximately

$$\Delta B_{out}^{wt} = [B(O2)\text{-}B(O1)] \approx C\ \{2\ [A_{\parallel}(^{14}N(5)) + A_{\parallel}(^{14}N(10))] + A_{z}(^{1}H(5))\} \qquad (2)$$

where $C = \frac{g_e\mu_B}{h} = 3.57\text{x}10^{-2}$ mT/MHz is a constant for translating the couplings observed in the spectrum (in mT) to MHz taking into account that the g-factor is close to the one of the free electron, $g \approx g_e$.

This, as it will be shown in the next section, can be used to estimate unknown large hyperfine couplings if the others are known or if a reference spectrum without the particular nucleus of interest is available.

### 3 Experimental results

### 3.1 Flavodoxin selectively labeled with $^{13}$C at positions 2 and 4 of the FMN ring

### 3.1.1 CW EPR

The X-band CW-EPR spectra for [$^{13}$C(2)-FMN]-Fld and [$^{13}$C(2, 4a)-FMN]-Fld samples are shown in Fig. 2. The hyperfine splittings of non-isotopically labelled Fld (WT Fld), described in a previous work (Martinez et al., 2016), is also shown here for comparison. According to what was said above, if the hyperfine interaction with the $^{13}$C nuclei in the labeled samples was large in the direction perpendicular to the isoalloxazine plane, the distance between the outermost O1 and O2 shoulders is predicted to increase. It can be seen that, for [$^{13}$C(2)-FMN]-Fld, the spectrum does not change with respect to the WT, nor

the O1-O2 distance, which indicates that the coupling of $^{13}$C nucleus at position 2 is small. On the other hand, an increase in the separation between the two outermost shoulders is clearly seen for [$^{13}$C(2, 4a)-FMN]-Fld. This indicates that the X-band CW-EPR experiments can provide a first estimate of the hyperfine splitting due to the $^{13}$C(4a) nucleus in the direction perpendicular to the isoalloxazine ring, since the broadening can be directly attributed to the hyperfine coupling of $^{13}$C(4a):



$$\Delta B_{\text{out}}^{13C(4a)FMN} \approx C \left\{ 2[A_z(^{14}N5) + A_z(^{14}N10)] + A_z(^1H5) + A_z(^{13}C4a) \right\} \qquad (3)$$

The differences between the B(O2)-B(O1) splitting of the WT Fld (or [$^{13}$C(2)-FMN]-Fld) and [$^{13}$C(2,4a)-FMN]-Fld samples are:

$$\Delta B_{\text{out}}^{13C(4a)FMN} - \Delta B_{\text{out}}^{wt} = 8.4 \pm 0.3 \text{ mT} - 7.1 \pm 0.3 \text{ mT} = 1.3 \pm 0.4 \text{ mT} \approx C\,A_z(^{13}C4a) \qquad (4),$$

which gives a first estimate of the hyperfine coupling in the direction perpendicular to the isoalloxazine plane of $A_z(^{13}C4a) \approx$ 36 MHz.

It should be noted that calculations of the hyperfine coupling of this nucleus in flavoproteins (Weber et al., 2001, García et
al., 2002) indicated that it is an anisotropic, nearly axial, interaction, with the largest splitting in the direction perpendicular to the plane. On the other hand, these calculations predicted a value for $A_\parallel(^{13}C4a)$ around double of that obtained in our experiment ($A_\parallel^{calc}(^{13}C4a) \approx$ 70-90 MHz). This discrepancy will be discussed later.

In order to gain accuracy and further information about the $^{13}$C(4a) hyperfine coupling, advanced EPR techniques were also used.

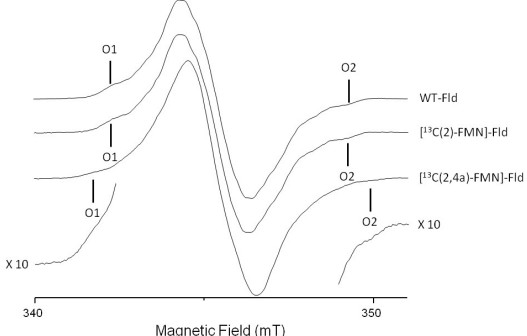


**Figure 2: X-band CW-EPR spectra of $^{13}$C labelled Fld variants at 50 K.**

### 3.1.2 ELDOR-detected NMR

The ELDOR-detected NMR technique can be used to detect nuclear frequencies in systems for which EPR transitions are
partially allowed due to, for example, hyperfine anisotropy and/or quadrupole interaction. Pumping an EPR forbidden transition burns a hole in the polarization that is detected by a decrement on the spin echo generated by the detection

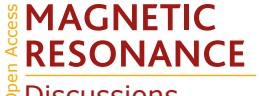

sequence at a variable frequency when it hits an allowed EPR transition. Negative peaks associated with nuclear frequencies are obtained when plotting the echo intensity as a function of the detection frequency (ELDOR frequency), at symmetric positions with respect to the detection frequency. Since this experiment is based on driving the polarization, the signal

obtained is free of blind spots and other distortions, although it has zero intensity for the principal directions of the hyperfine tensor (since in those directions, EPR transitions are completely allowed).

Figure 3 shows the ELDOR detected NMR experiments of [$^{13}$C(2)-FMN]-Fld and [$^{13}$C(2, 4a)-FMN]-Fld. For the $^{13}$C(4a) nucleus, the values of the hyperfine coupling and the Larmor frequency at Q-band (13.0 MHz) are comparable. Therefore, Q-band ELDOR-detected NMR is especially suitable for detecting nuclear frequencies of $^{13}$C(4a). The experiments were

performed with the magnetic field set to the EPR absorption maximum (Fig. 3.b), as well as on the tails of the CW spectrum (Fig. 3.a). As mentioned before, in the second case orientation selection occurs since only molecules oriented with the isoalloxazine plane approximately perpendicular to the direction of the magnetic field will contribute to the spectrum (Martínez et al., 2014). On the other hand, experiments recorded of the magnetic field at the center of the spectrum are contributed by all possible orientations in the disordered sample (see insets in Fig. 3). The spectra obtained by the

subtraction of the two $^{13}$C labelled samples measured under identical conditions is also shown for both magnetic field positions and the obtained lines correspond to the nuclear frequencies of the $^{13}$C(4a) nucleus (Fig. 3, bottom spectra).

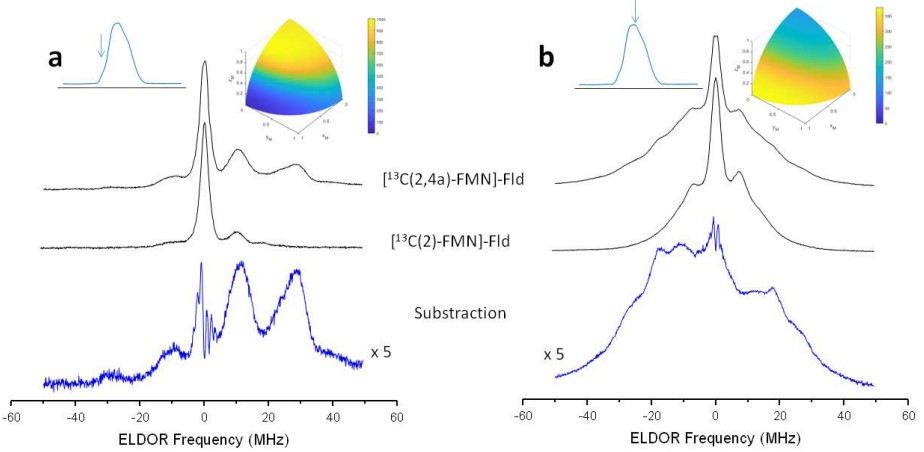

**Figure 3: ELDOR-detected NMR spectra of $^{13}$C labelled Fld variants**. a) Spectra taken at the low-field tail of the CW spectrum, B = 1221.4 mT for [$^{13}$C(2,4a)-FMN]-Fld and B = 1222.2 mT for [$^{13}$C(2)-FMN]-Fld, both corresponding to selective excitation of molecules with the magnetic field oriented perpendicular to the isoalloxazine ring (A$_{//}$ since the magnetic field is parallel to the axis). b) Spectra taken at the maximum absorption of the CW spectrum, B = 1225 mT for both samples, which yields no orientation selection. For all spectra, T = 50 K. The upper inserts on the left shows the Echo-detected EPR spectrum of the sample with the magnetic field setting of the experiment





indicated by an arrow. The inserts on the right shows the pattern of excited orientations in a sphere octave according to the colours of the accompanying scale.

For the tail spectra, two wide signals are obtained, respectively centered around ± 30 MHz and ± 11 MHz. The value of the largest one is larger than $2\nu_L(^{13}\text{C}) = 13.1$ MHz, which indicates that the nucleus is in the strong coupling regimen at this

particular orientation ($|A_z| > 2\nu_L(^{13}\text{C})$). In this regimen, nuclear frequencies for a nucleus with I = 1/2 are approximately

$$\nu_+ = \left|\frac{A_z}{2} + \nu_L\right| \;,\;\; \nu_- = \left|\frac{A_z}{2} - \nu_L\right| \tag{5}$$

and therefore

$$\nu_+ - \nu_- = 2\nu_L \;,\;\; \nu_+ + \nu_- = |A_z| \tag{6}$$

The difference between the frequency of the detected peaks is close to $2\nu_L(^{13}\text{C})$, which confirms the assignment of the peaks

to the nuclear frequencies of $^{13}$C4a. On the other hand, since the selection by field is not perfect (see insets in Fig. 3), the peaks in the spectrum are contributed by molecules for which the direction perpendicular to the isoalloxazine plane presents a wide distribution of orientations around that of the magnetic field. It should be also taken into account that the direction perpendicular to the isoalloxazine plane ($z$) is likely one of the principal axes of the hyperfine tensor of this nucleus (García et al., 2002; Weber et al., 2001), so molecules perfectly aligned with the field do not contribute to the ELDOR detected

NMR spectra. The edge of the detected broad signals would then be a first estimation of the value for the nuclear transitions in this direction $|A_z[^{13}\text{C}(4a)]| \approx 40$ MHz, compatible with the estimations from X-band CW-EPR experiments. The signals being wide indicate that the hyperfine tensor of $^{13}$C(4a) is quite anisotropic.

In the experiment at the center field very wide signals are also distinguished, as expected for an orientationally disordered sample. The main distinct feature of the spectrum is a peak at ± 18 MHz. It could correspond to the perpendicular feature of

the larger nuclear frequency since many of the orientations are close to the perpendicular plane. In such a case, its absolute value would be approximately $|A_{x,y}[^{13}\text{C}(4a)]| \approx 10$ MHz. Assuming that the hyperfine tensor is very anisotropic, it is likely that $A_z[^{13}\text{C}(4a)]$ and $A_{x,y}^{13}\text{C}(4a)]$ will exhibit opposite signs. The evidence from the ELDOR detected NMR experiments does not allow resolving $A_x$ and $A_y$ of $^{13}$C(4a) but since the hyperfine interactions in the flavin ring tend to be nearly axial, we can assume that $A_x[^{13}\text{C}(4a)]$ would be close to $A_y[^{13}\text{C}(4a)]$. Then a first estimation from the analysis of these experiments would

be:

$$A_z[^{13}\text{C}(4a)] \approx +40 \text{ MHz} \qquad\qquad A_\perp[^{13}\text{C}(4a)] \approx -10 \text{ MHz}$$

These values can be further refined from evidence obtained from Q-band HYSCORE experiments



### 3.1.2 HYSCORE

Q-band HYSCORE experiments were also carried out in [$^{13}$C(2)-FMN]-Fld and [$^{13}$C(2,4a)-FMN]-Fld samples, both at the center (Fig. 4) and at the tail of the EPR line (Supplementary Material). Again, the larger Larmor frequency of $^{13}$C at Q-band allows some weak features from echo modulation with $^{13}$C(4a) nuclear frequencies to be seen in the spectra.

The HYSCORE spectra of both samples show intense ridges in the negative quadrant due to $^{14}$N that will be discussed later. Since the focus here is on the $^{13}$C signals, Fig. 4 only shows the positive quadrant of the 2D measurement, set at the center of

the EPR line. The spectrum of [$^{13}$C(2)-FMN]-Fld, on the right, shows a small elongated ridge on the antidiagonal that crosses the diagonal at the $^{13}$C Larmor frequency. This line is attributed to the hyperfine coupling of $^{13}$C(2), which was estimated smaller than 2 MHz. The low frequency correlations are assigned to $^{14}$N. The spectrum of [$^{13}$C(2,4)-FMN]-Fld is shown in the center and besides the described lines, a weak and broad ribbon-shaped symmetric feature is distinguished on the diagonal, with the knot at about 17 MHz. The long streamer of this ribbon reaches the $^{13}$C antidiagonal at about (6, 20) MHz

and a shorter inner ridge is distinguished almost perpendicular to the axis. We interpret this feature, undoubtedly associated to $^{13}$C(4a), to be the crossing part of two long ridges starting (although not visible) at (40,11) MHz, point corresponding to $z$, the orientation perpendicular to the isoalloxazine plane. The rhombicity of the hyperfine tensor within the plane is manifested by the width of the feature (red area in Fig. 4.c), especially in the two ridges reaching to the antidiagonal which are the outer borders of the "ribbon". The points where these structures cross the antidiagonal allow estimating the two

principal values of the hyperfine tensor in the flavin plane and confirm the hyperfine couplings in the plane and the hyperfine coupling perpendicular to the plane have opposite signs.

Using the value for $A_z$ estimated from the analysis of the ELDOR detected NMR experiments, the following values are obtained from the analysis and simulation of the spectra for the three principal values of the hyperfine tensor:

$A_z[^{13}C(4a)] = (+40 \pm 2)$ MHz        $A_2[^{13}C(4a)] = (-13.5 \pm 1)$ MHz        $|A_3[^{13}C(4a)]| = (-9 \pm 1)$ MHz


The hyperfine couplings within the isoalloxazine plane could not be assigned to particular directions in the plane, therefore, they have been labeled $A_2$ and $A_3$ instead of $A_y$ and $A_x$. The solid lines superimposed to the spectra on the right spectrum are the HYSCORE patterns calculated with the couplings given above. Proper simulations of spectral intensities are shown in the Supplementary Material.



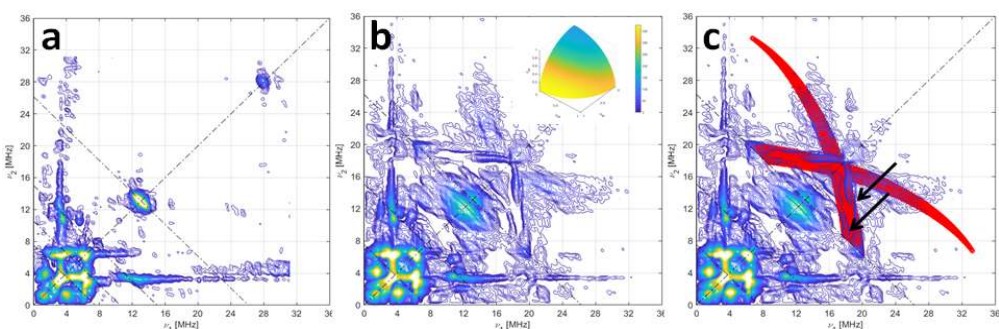


**Figure 4: HYSCORE of $^{13}$C labelled Fld variants**. a) [$^{13}$C(2)-FMN]-Fld spectrum, b) [$^{13}$C(2,4a)-FMN]-Fld spectrum, and c) [$^{13}$C(2,4a)-FMN]-Fld spectrum with the calculated HYSCORE pattern for a $^{13}$C nucleus with the parameters specified in the text superimposed in red. Both experimental spectra were taken at the maximum absorption of the CW-EPR spectrum corresponding to no orientation selection, B = 1226.5 mT, $\tau$ = 112 ns and T = 50 K. Antidiagonal lines cross the diagonal at the Larmor frequencies $\nu_{14N}$, $2 \cdot \nu_{14N}$ and $\nu_{13C}$. The two arrows

on the right point the two ridges that evidence the rhombicity of the $^{13}$C(4a) hyperfine tensor. The inset in b) shows the orientation selection of the experimental spectra in a sphere octave according to the colours of the accompanying scale.

## 3.2 Flavodoxin isotopically labeled with $^{15}$N at the FMN ring

### 3.2.1 HYSCORE

Although the Q-band HYSCORE spectra of samples with natural abundance of nitrogen nuclei present intense signals due to hyperfine interactions with these nuclei, their interpretation is difficult, because the $^{14}$N nucleus has a nuclear spin $I = 1$ and an appreciable quadrupole contribution, which causes the appearance of multiple correlation features (see Fig. 6). The use of samples labelled with $^{15}$N-FMN greatly simplifies the analysis, since its $I = 1/2$ nucleus presents a single nuclear transition per electron spin manifold, and therefore a single pair of correlated features per nucleus. The hyperfine parameters obtained

from $^{15}$N are directly convertible to those of the $^{14}$N nucleus at the same position.

**a**



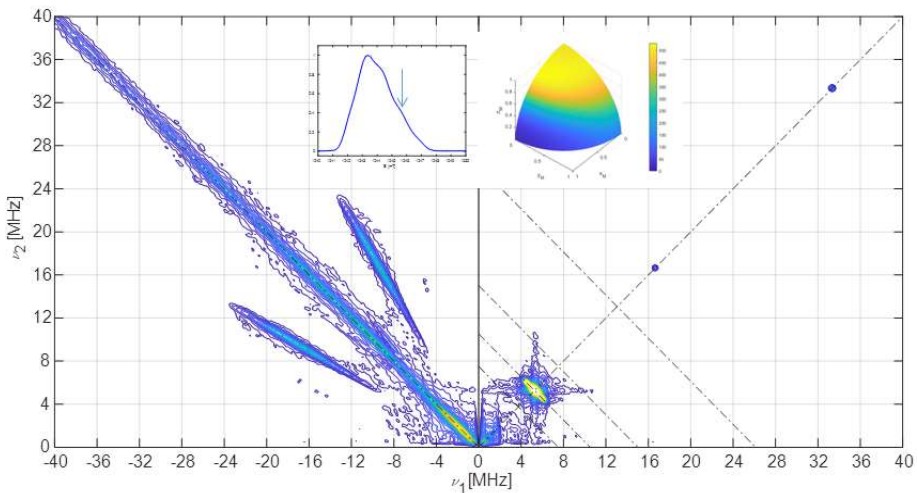

**b**

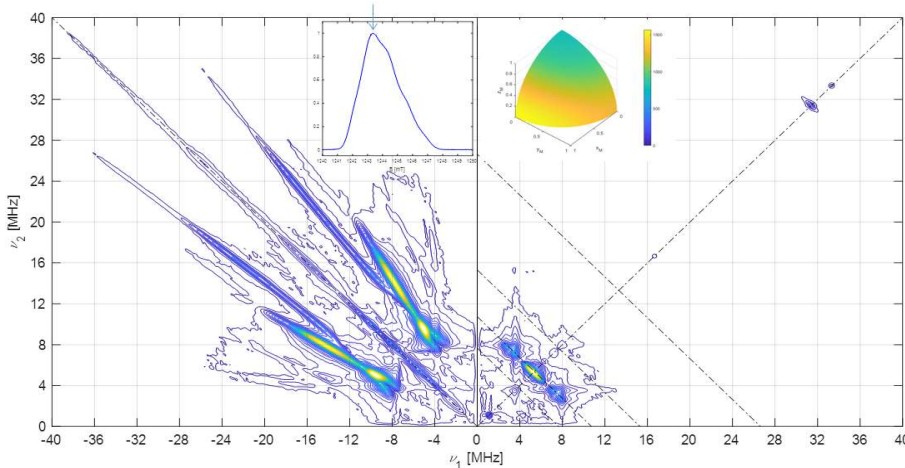

**Figure 5: HYSCORE of [$^{15}$N-FMN]-Fld**. a) Spectrum taken at the high-field tail of the CW spectrum, B = 1219.7 mT, the displayed spectrum is the sum of spectra taken at τ values of 96, 124, 144 and 168 ns. b) Spectrum taken at the absorption maximum of the CW spectrum, B = 1217.2 mT, the displayed spectrum is the sum of spectra taken at τ values of 96, 144 and 168 ns. T = 80 K for all spectra. The antidiagonal line crossing the (+,+) diagonal at the Larmor frequency ν$_{15N}$ has been included for reference. The upper insert on the left shows the Echo-detected EPR spectrum of the sample with the magnetic field setting of the experiment indicated by an arrow. The insert on the right shows the pattern of excited orientations for each spectrum.



The Q-band HYSCORE experiments performed in a [$^{15}$N-FMN]-Fld sample are displayed in Fig. 5, both, at the upper tail of the EPR spectrum (Fig. 5.a) and at its maximum (Fig. 5.b). To the first spectrum, only molecules with the magnetic field approximately perpendicular to the flavin plane contribute (see insert in Fig. 5.a). The hyperfine coupling of N(5) at this

orientation is too large (Martínez et al., 1997; Weber et al., 2005) for the excitation bandwidth of the mw pulses to be enough to excite its nuclear frequencies. The spectrum, therefore, only shows a pair of ridges, symmetrical with respect to the diagonal and approximately parallel to it. This features can be associated with a hyperfine interaction with a nucleus of spin I = ½ and attributed to N(10). From the distance between these ridges, the hyperfine interaction of N(10) close to $z$ can be estimated. The short ridge on the $^{15}$N antidiagonal in the positive quadrant is assigned to weakly interacting nuclei N(1) and

N(3) whose hyperfine couplings have been reported somewhere else (Martínez et al. 2012).

The HSYCORE spectrum at the maximum of the EPR line (Fig. 5.b) displays two crossing pairs of correlated ridges, one of them is very long, associated with a highly anisotropic and strong interaction, and assigned to N(5) according to previous results (Martínez et al., 1997; Weber et al., 2005). The other is shorter and overlaps partially with the ridge seen in the parallel spectrum and is assigned to N(10). Additionally, two pairs of peaks on the $^{15}$N antidiagonal come out in the positive

quadrant at the low-frequency edge of the two ridges, allowing to identify the hyperfine coupling in the plane as isotropic within the plane. Satisfactory simulations (see Supplementary Material) were produced using the following axial hyperfine parameters:

$A_z[^{15}N(5)] = (+74 \pm 3)$ MHz $\qquad A_\perp[^{15}N(5)] = (+5.6 \pm 0.3)$MHz

$A_z[^{15}N(10)] = (+38.0 \pm 1.0)$ MHz $\qquad A_\perp[^{15}N(10)] = (+3.2 \pm 0.3)$MHz


The orientation of the tensor cannot be obtained from our results, the one published by Kay and coworkers was used (Fuchs et al., 2002, Kay et al., 2005) for the simulations.

### 3.3 HYSCORE signals of $^{14}$N at the FMN ring

Once the hyperfine coupling parameters of $^{15}$N are estimated with high precision, they can be directly transformed to the ones of $^{14}$N in the same position by just applying a factor $g_N(^{14}N)/g_N(^{15}N) = -0.71$. With the hyperfine already established, the experimental data on the $^{14}$N-FMN can be used to refine the values of the quadrupole tensor. In Fig. 6, the complete set of spectra for [$^{13}$C(2)-FMN]-Fld are shown. As mentioned before, they are dominated by $^{14}$N ridges. The spectrum at the high-field tail shows two short pairs of ridges assigned to N(10) for the reason mentioned above. It allows obtaining a value

of 0.8 MHz for $|Q_z|$, the principal value of the quadrupole tensor in the direction perpendicular to the isoalloxazine ring. The spectrum recorded at the center of the EPR line contains ridges due to N(10) and N(5) (Fig. 6.b). The best simulation of the spectra, shown in Fig. S.6 of the Supplementary Material, was produced with the following quadrupole parameters:

$Q_z[^{14}N(10)] = -0.8$ MHz $\qquad Q_{\perp,1}[^{14}N(10)] = 2.4$ MHz $\qquad Q_{\perp,2}[^{14}N(10)] = -1.6$ MHz

$Q_z[^{14}N(5)] = 1.8$ MHz $\qquad Q_{\perp,1}[^{14}N(5)] = -0.8$ MHz $\qquad Q_{\perp,2}[^{14}N(5)] = -1.0$ MHz



With our data, these principal values cannot be associated with a particular axis in the plane and we have, therefore, labeled the principal values in the flavin plane as $Q_{\perp,1\ or\ 2}$.

**a**

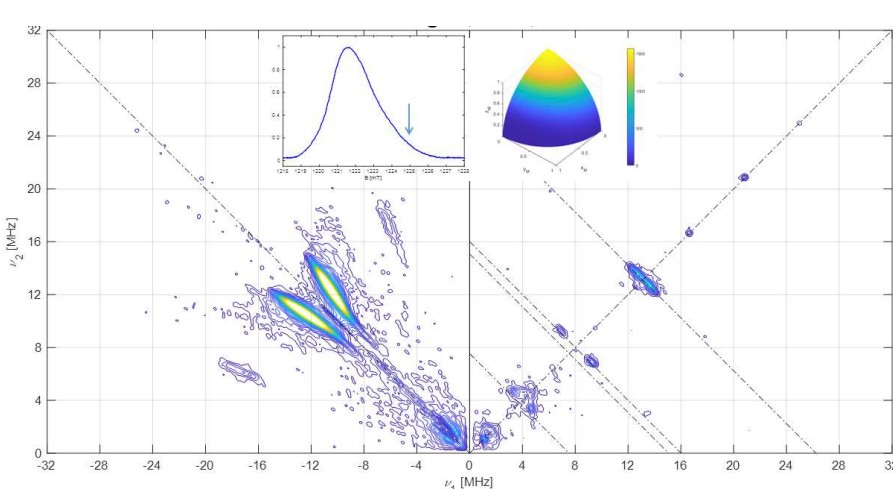

**b**

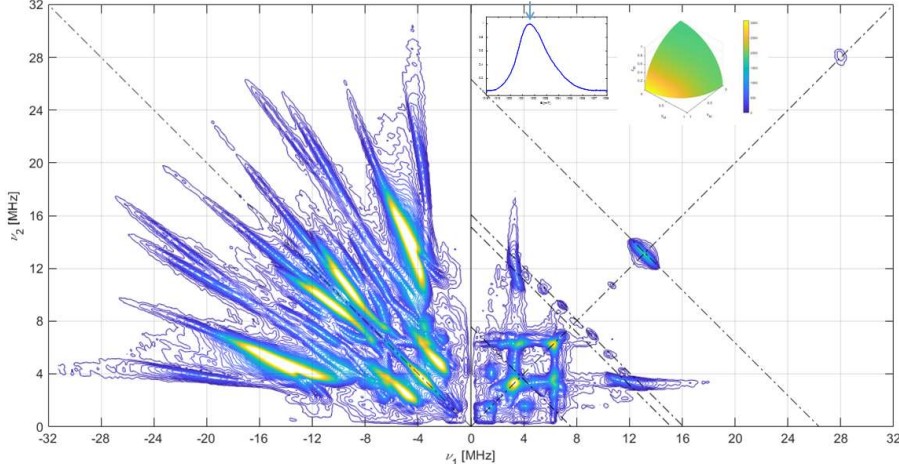



**Figure 6: $^{14}$N HYSCORE signals Fld samples with naturally abundant nitrogen in the FMN.** Spectra recorded using the sample [$^{13}$C(2)-FMN]-Fld. a) Spectrum taken at the tail of the CW spectrum, B = 1225.0 mT, the displayed spectrum is the sum of spectra taken at τ values of 96, 112, 128, 144 and 176 ns, b) Spectrum taken at the absorption maximum of the CW spectrum, B = 1221.0 mT, the displayed spectrum is the sum of spectra taken at τ values of 96, 128 and 208 ns. T = 50 K for all spectra. The antidiagonal lines crossing the (+,+) diagonal at the Larmor frequencies $\nu_{14N}$, $2\cdot\nu_{14N}$, $\nu_{2H}$ and $\nu_{13C}$ have been included for reference. The upper insert on the left shows the Echo-detected EPR spectrum of the sample with the magnetic field setting of the experiment indicated by an arrow. The insert on the right shows the pattern of excited orientations for each spectrum.

## 3 Discussion

In this work, the use of hyperfine spectroscopy techniques in Q-band, in combination with selective isotopic labeling, have proven very useful to experimentally determine the hyperfine couplings of the unpaired electron of flavoproteins in the semiquinone state with the nuclei of the isoaloxacine ring bearing the highest spin densities. The complete set of principal hyperfine values of $^{13}$C(4a), $^{15}$N(5) and $^{15}$N(10) nuclei have been determined. Additionally, the spectroscopic effects of the nuclear quadrupole interaction of $^{14}$N(5) and $^{14}$N(10) have been analyzed. Despite the relevance of these positions in the ring, until now only partial experimental evidence of hyperfine interactions was available. The interaction with $^{15}$N(10) and $^{14}$N(10) in this protein had been previously determined by us using X-band HYSCORE experiments (Martínez et al., 1997). The Q-band results presented here show that both, hyperfine and quadrupolar interactions are compatible with and allow refinement of the ones previously reported. Concerning N(5), the hyperfine structure was never obtained for this protein but the hyperfine matrix was determined by an analysis of the X-band and W-band CW-EPR spectra assisted by simulation in the flavoprotein Na$^+$-NQR (Barquera et al., 2003). Here we have shown that direct evidence for the hyperfine interaction with N(5) can also be obtained using Q-band HYSCORE experiments providing very similar results in fld. Estimations for the quadrupolar interaction for this nucleus were also obtained for the first time due to the high resolution of 2D HYSCORE experiments. In spite of the symmetric positions of N(5) and N(10) in the pyrazine central ring, the nuclear quadrupole values differ considerably, possibly indicating a significant contribution of the unpaired electron density, considerably larger in the former atom.

Regarding the hyperfine interaction with $^{13}$C, our X-band CW-EPR, ELDOR detected NMR and Q-band HYSCORE experiments allowed us to determine the complete set of the principal values of the hyperfine interaction. Previously, it was only possible to determine for DNA photolyase the two smallest principal values (Schleicher et al., 2021), which are in the same range of the ones found here. For the hyperfine couplings of $^{13}$C(2), we find also compatible values for fld, always smaller than 2 MHz. It is worth noting that only the complete characterization of the hyperfine structure allows obtaining the isotropic and anisotropic components of $^{13}$C(4a), see Table 1, the former related to the spin density at the nucleus and the second one directly related to the spin population in the π-orbitals of the atom. In addition, accurate values of the largest hyperfine splittings in the isoalloxazine ring are critical to characterize the magnetochemistry involved in the



magnetoreception of avian cryptochromes (Hore et al., 2016), so the reported experimental values of should be very useful for modelling this mechanism.

For other positions in the ring, experimental evidence shows only small differences in the hyperfine structure between different flavoproteins exhibiting neutral semiquinone, even when their functions are very different. It remains to be proved whether this holds true for the newly estimated spin densities of C(4a) or they vary among the different flavoproteins.


Table 1.- Comparison between measured and calculated hyperfine parameters for the flavin ring sites with the largest hyperfine couplings.

| | Measured Isotropic hyperfine parameter $a$ (MHz)[a] | Calculated Isotropic hyperfine parameter $a$ (MHz)[a] | Relative inaccuracy in $a$ (exp-calc, %) | Measured anisotropic hyperfine parameters $T_x$, $T_y$, $T_z$[b] (MHz) | Calculated anisotropic hyperfine parameters $T_1$, $T_2$, $T_3$[b] (MHz) |
|---|---|---|---|---|---|
| $^{13}$C(4)[c] | -9.7 | -11.2 | -16% | -4.1, -1.5, +5.7 | -0.4, -1.3, +1.8 |
| $^{13}$C(4a) | +5.8[d] | +13.2[c] | 126% | -14.8, -19.4, +34.2[d] | -23.3, -22.7, +46.0[c] |
| $^{14}$N(5) | +20.2[d,e] | +13.6[f] | -33% | -16.2, -16.2, +32.4[d,e] | -14.6, -14.6, +29.2[f] |
| $^{13}$C(5a)[c] | -13.2 | -12.3 | 7% | -1.9, +0.2, +1.6 | +1.7, +2.3, -3.9 |
| $^{14}$N(10) | +10.5[d,e] | +7.6[f] | -28% | -8.2, -8.2, +16.5[,e] | -7.6, -7.6, +15.2[f] |
| $^{13}$C(10a)[c] | -14.0 | -13.6 | 3% | -1.4, +0.4, +0.9 | +1.1, -0.7, -0.5 |

*a*. Isotropic hyperfine parameter $a = \frac{(A_x + A_y + A_z)}{3}$

    *b*. $T_i = A_i - a$, $i = x, y, z$

    *c*. Ref. [Schleicher, 2021]

    *d*. This work, in Fld

    *e*. The hyperfine parameters for $^{14}$N nucleus have been scaled from $^{15}$N ones (see text).

*f*. Ref. [García, 2002]

When a comparison is made with previously calculated values of the hyperfine interaction, our results are quite surprising.

Given that all hyperfine interactions had been estimated by calculations, and considering the good general agreement that existed with the known experimental values, until now the values calculated also for the $^{13}$C(4a) nucleus were considered





plausible. Nevertheless, our results show that the hyperfine interaction of $^{13}C(4a)$ predicted by calculations is significantly overestimated, regardless of whether its isotropic or anisotropic part are compared. This overestimation represents the most significant discrepancy within the flavin isoalloxazine ring in percentage terms. On the other hand, this could be related to

the previously mentioned underestimation of the hyperfine splitting in the $^{14}N(5)$ and $^{14}N(10)$ nuclei. The discrepancies between calculations and experimental values could, therefore, indicate that this issue is not specific of type of nucleus for which the interaction is estimated. Instead, it may reflect a broader difficulty in the ability of the of the calculations to predict realistic hyperfine interactions or the potential overlooking of a relevant factor. It is worth noting that reported calculations took both actual flavin structures in the semiquinone state from X-ray diffraction experiments and DFT

optimized structures, obtaining very similar results.

Table 1 shows the comparison between experiments and calculations for the ring nuclei with the highest hyperfine couplings (4, 4a, 5, 5a, 10, 10a), where both the isotropic and the anisotropic parts of the hyperfine principal values are considered. The values of the isotropic hyperfine parameter calculated for all $^{13}C$ nuclei, except for position 4a, reproduce the real values quite well. Concerning the anisotropic part, it is relatively small in all these nuclei, and has a marked orthorhombic character,

which indicates an important mixture of $\sigma$ orbitals in the SOMO for those positions. Besides, the isotropic hyperfine constants obtained in the calculations are clearly underestimated for $^{14}N(5)$ and $^{14}N(10)$, and severely overestimated for $^{13}C(4a)$, in which a value more than double the one obtained experimentally is predicted. Furthermore, a similar trend occurs when comparing the calculated and measured data for the anisotropic part of the interactions. The calculations reproduce well the almost axial character of the hyperfine matrices for the three nuclei, but underestimate their magnitude for $^{14}N(5)$

and $^{14}N(10)$ (between 10% and 8%), and overestimate it for $^{13}C(4a)$ (around 35%). The overestimation in the $^{13}C(4a)$ calculations is quite significant, while in the calculations the interactions with the carbon nuclei in positions 4 and 5a are comparable with those in position 4a (and in previous calculations they were considered much smaller, see Weber et al., 2001), the experimental values reveal that the latter has a value nearly half that of the others. Regarding the anisotropic part, the one of $^{13}C(4a)$ remains the largest among the carbon nuclei, but its experimental value makes it comparable to that of the

$^{14}N(5)$ and $^{14}N(10)$ nuclei, despite the fact that the nuclear gyromagnetic factor in these nuclei is quite smaller than that of $^{13}C$.

All this shows that the current calculations present an essential difficulty in realistically describing the electronic spin distribution of SOMO, which in turn could indicate the need to also improve other aspects of electronic structure prediction. Our findings suggest that the spin density predicted at position 4a could be actually shifted towards the central positions of

pyrazine (5 and 10), which may have important consequences on the understanding of the electron transfer mechanisms that specifically involve these positions. The fact that the only significant differences between the hyperfine calculations and the experimental values relate to a shift of the electron density between the two of the most reactive positions of the flavin cofactor is intriguing and it remains to be investigated whether this is a general feature in the semiquinone state of flavoproteins or it can be hypothetically associated to a modulation of the reactivity by the protein environment. For

example, more electron density in position C(4a) would favor reoxidation whereas more electron density in N(5) would



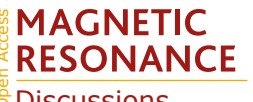

promote hydride transfer. Whether this modulation is due to a specific protein-flavin interaction or a structural distortion of the isoalloxacite ring is certainly interesting and remains to be identified in future studies.

***Supplementary Material*** The supplementary Material is available together with the online version of this article.

***Author contributions*** JIM and MM designed the research. SF prepared the protein samples. IGR carried out the experiments and analyzed the data together with JIM. JIM, IGR and MM wrote the manuscript.

***Acknowledgments***

Riboflavin analogues were a generous gift from Dr. D. Edmondson. Gunnar Jeschke kindly provided access to the spectrometer park of his laboratory to perform the experiments

***Financial support***

This work has been funded by the Spanish State Research Agency and by FEDER (MCIN/AEI-FEDER, Grants PID2022-136369NB-I00, PID2022-140923NB-C21 and PID2021-127287NB-I00) and the Regional Government of Aragón-FEDER (grants E35_23R and E09-23R), as well as MCIN with resources from European Union NextGenerationEU (PRTR-C17.I1) promoted by the Government of Aragon

***Competing interests.*** The authors declare no conflict of interest.

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
