# Peer review of "Determining large hyperfine interactions of a model flavoprotein in the semiquinone state by pulse-EPR techniques"

_Magnetic Resonance, 2024_

## Referee Comment (RC2)

**Specific comments**

Materials and Methods

- Line 177: *A phase cycle of eight steps.*
  More than one 8-step phase cycling scheme has been proposed for 4-pulse HYSCORE, see *e.g.* DOI 10.1016/0022-2364(90)90181-8 and DOI 10.1007/s00723-008-0140-6.
  It would be beneficial to highlight which one was used and add the corresponding reference.
- Lines 183-184: *The pulse lengths were 1 μs for the first variable mw frequency ($mw_2$) [...].*
  - What was the amplitude of the HTA pulse $\omega_{HTA}$ measured at the observer frequency ($mw_1$)?
  - Was such a short HTA pulse used indeed?
- Line 207: which specific reference was used for the listed **g** principal values?

ELDOR-detected NMR

- Line 261: the text mentions the variation of the *spin echo* intensity, however FID-detected EDNMR has been used for this work.
- Lines 261-262: *is detected by a decrement on the spin echo generated by the detection sequence at a variable frequency when it hits an allowed EPR transition*.
  - It may be better to rewrite this sentence as it seems as if the observer frequency $mw_1$ were swept during the experiment.
  - Also line 263: *are obtained when plotting the echo intensity as a function of the detection frequency (ELDOR frequency)*; this is not in agreement with what is stated in 2.2.3, *first pulse with variable frequency ($mw_2$) [...] second pulse with fixed mw frequency $mw_1$*.
- Line 269: *Q-band ELDOR-detected NMR is especially suitable for detecting nuclear frequencies of $^{13}C(4a)$*.
  - Some clarification of why this is the case would be beneficial, especially considering the resolution limitations of EDNMR compared to ENDOR.
  - Related to this point, according to the cited references for the spectrometer (Gromov *et al.*, 2001) and the microwave resonator (Tschaggelar *et al.*, 2009) the used equipment should be capable of performing ENDOR measurements. Was there a specific reason to choose EDNMR over Davies ENDOR? Was Davies ENDOR attempted on the studied system?
- Figure 3:
  - The EDNMR spectra are strongly asymmetric. This may impact the quality of the subtraction. How was the experiment set up? Where was the detection frequency placed within the resonator mode?
  - Panel a: the signals at 11 MHz appear in the spectra of both [$^{13}C(2)$-FMN]-Fld and [$^{13}C(2,4a)$-FMN]-Fld.
    Is the presence of a signal at 11 MHz in the subtraction spectrum an artefact caused by different acquisition conditions (*e.g.*, different $\omega_{HTA}$ field strength, resonator bandwidth, position of the pulses in the resonator mode) between the two experiments?
  - Related to the comment about the *Materials and Methods* section, is the width of the central blind spot compatible with a 1 μs HTA pulse?

- The *x*-axis of both panels reads *ELDOR frequency (MHz)*.
  This nomenclature may be misleading: the spectra are displayed against the frequency difference between the HTA pulse and the detection frequency; the frequency of the ELDOR source should be in the ~34 GHz range.
- Line 283 (legend): *which yields no orientation selection*. The figure inset shows indeed some orientation selection; it may be better to use a milder statement, *e.g.* "negligible" or "weak".
- Line 285 (legend): how were the orientation selection spheres obtained? Which spin Hamiltonian was used? Which pulse excitation bandwidth?

- Line 289: *$2v_L(^{13}C) = 13.1$ MHz*. $v_L(^{13}C)$ at the chosen magnetic field should be ~13.11 MHz, hence $2v_L(^{13}C)$ is ~26 MHz.
- Line 294: *The difference between the frequency of the detected peaks is close to $2v_L(^{13}C)$, which confirms the assignment of the peaks [...]*.
  The frequency difference between the peaks, located at ±30 MHz and ±11 MHz, is approximately 19 MHz.
  This value is rather far from $2v_L(^{13}C) = 26$ MHz.
- Lines 295-297: I would consider rewriting the sentence as, in its current form, it may lack some clarity.
- Lines 300-301: it's not entirely clear to me how the edge of the outermost signal in the EDNMR spectrum could be converted directly into a value of $|A_z|$, especially considering that the system is in the strong-coupling case. Was equation 6, $v_+ + v_- = |A_z|$, used to estimate $|A_z|$?
- Line 302: I agree with the observation, especially for the outermost signal.
  Were measurements performed at several $\omega_{HTA}$ field strengths and/or different lengths of the HTA pulse to ensure the absence of additional broadening due to the choice of the experimental conditions?
- Line 305: is the notion *larger nuclear frequency* related to $v_+$?

**$^{13}C$ HYSCORE**

- The full 2D spectra (both (++) and (-+) quadrants) for [$^{13}C$(2,4a)-FMN]-Fld are displayed neither in the main text nor in the SI.
  - I find this quite surprising, especially considering that the hyperfine interaction with $^{13}C$(4a) is expected to be in the strong coupling regime and should hence give signals in the (-+) quadrant as well.
  - Is the (-+) region of the mentioned spectrum devoid of any signals?
- Line 319: *Since the focus here is on the $^{13}C$ signals, Fig. 4 only shows the positive quadrant of the 2D measurements*; see the comment above.
- As $I(^{13}C) = ½$, it should be possible to obtain an initial estimate of $a_{iso}$ and $|T|$ by squaring both frequency axes, see DOI 10.1006/jmra.1995.1199.
- Line 334: $|A_3[^{13}C$(4a)]|$ is reported to be negative. Is the absolute value a typo?
- Line 334: how were the uncertainties estimated?
- Line 339 (and several other places throughout the manuscript): it would be better to have specific references to figures in the Supplementary Material.
- Figure S.1: it would be better to remind the readers of the field position at which the experiment was performed.

- Figure S.2: it would be better to display the precise location with respect to the EDFS-EPR spectrum at which the experiment was performed. Is it *e.g.* the same as the one in Fig. 3a or a different one?

$^{15}$N and $^{14}$N HYSCORE

- Figures 5 and 6
  - Is there a specific reason for performing the field-edge $^{15}$N and $^{14}$N experiments on the high-field side of the spectrum rather than on the low-field side? This latter position was *e.g.* chosen for the EDNMR experiments (see Fig. 3a) and for the field-edge $^{13}$C-HYSCORE measurement (see Fig. S2).
  - There is some conflict between the main text and the SI: according to the legend of Fig. S5, the experiment would have been performed at the low-field edge of the EPR spectrum whereas according to the corresponding Fig. 6a the measurements were performed on the high-field edge.
  - The field settings are rather different between Fig. 5a ($^{15}$N) and Fig. 6a ($^{14}$N). According to which criterion were the edge positions selected?
  - In the legends of Figures 5 and 6 it is mentioned that HYSCORE spectra were recorded at several τ values and summed. How was this performed exactly? Were there major echo intensity changes between the shortest and the longest τ values?
  - Figure 5: at least 4 dash-dot antidiagonal lines are reported. Which nuclei do these correspond to? Considering that the focus is on $^{15}$N, it may be better to highlight the corresponding Larmor frequency.
  - Figures 6, S.5 and S.6: why are antidiagonal lines at the $^{2}$H Larmor frequency reported? Were the samples partially deuterated?
- Lines 383-384: how were the uncertainties estimated?
  How could some rhombicity of the $^{15}$N hyperfine coupling tensors be completely ruled out from the analysis of the experimental data?

General remarks

- I couldn't find any information concerning the availability of the raw data.
- The limited resolution of the figures makes it sometimes challenging to observe the details described and discussed in the text.

**Technical corrections**

- Line 98: *though → through?
- Lines 133-134: *These results provide with a suitable protocol to experimentally access these coupling and an estimation of the spin density on the Fld model*. This sentence is not entirely clear to me.
- Line 146: *at al. → et al.
- Line 168: although clear to a readership with magnetic resonance background, it may be better to expand the abbreviation *mw* the first time it is used.
- Line 203, equation (1): the hyperfine term does not carry the sum symbol ($\Sigma_i$).
- Line 205: *semiquinome → semiquinone*.
- Line 212: *use to be (subject/verb agreement).
- Line 218: *isoaloxacine; here, as well as in a few other places, the term *isoalloxazine* is spelt incorrectly.
- Line 228: **4a** instead of ***4**.
- Line 240, equation (3): the form is not consistent with equation (2) ($A_{\parallel}(^{14}N5)$ in equation 2 vs. $A_z(^{14}N5)$ in equation 3).
- Line 245, equation (4): for the sake of better readability, it would be better to enclose values and the related uncertainties within brackets, *i.e.* (8.4 ± 0.3 mT) - (7.1 ± 0.3 mT).
- Line 261: *on the → of the*.
- Figure 3, text between panels *a* and *b*: *Substraction → Subtraction.
- Line 290: *regimen → regime.
- Line 294: $^{13}C4a$ → $^{13}C(4a)$.
- Line 307: *$A_{x,y}^{13}C(4a)]$ → $A_{x,y}[^{13}C(4a)]$.
- Line 315: *carried out in → carried out on.
- Line 367: *performed in a → performed on a.
- Line 372: *symmetrical with respect to the diagonal* or antidiagonal?
- Line 376: *HSYCORE → HYSCORE.
- Line 416: *in Q-band → at Q-band.
- Line 416: *have → has.
- Line 423: * both, hyperfine → both hyperfine.
- Line 427: *fld → Fld.
- Line 432: for the sake of completeness, it may be better to stress *Q-band ELDOR-detected NMR*.
- Line 435: *fld → Fld.
- Line 466: *specific of type → specific of the type.
- Line 467: *of the of the → of the.

---

## Author Comment (AC6)

REFEREE #1

The authors have adressed most points and improved the manuscript, yet some things still need to be clarified.

Orientation Selection:

• No simulation is provided in Figure1 as stated by the authors
In Figure 1, the orientation of the g frame is shown but no simulation is intended. This orientation was reported by Kay et al in their paper of 2005 and it is the orientation that has been used in the simulations that are shown in other figures of the paper and the supplementary information. The wording in the figure caption of Figure 1 was changed to make this point more clear.

• The orientation selection map is only informative if the simulation of the Q band spectra is accurate, which cannot be judged here

The simulation of the Q-band EPR absorption spectra has been added to the inserts of the figures in the new version of the manuscript.

EDNMR:
• The accuracy of the frequencies given in the EDNMR figure is highly questionable given the broad features in the experimental spectra
The referee is right, the experimental features are broad and the frequency values given in the figure are subjected to some uncertainty. Therefore, we decided to show the frequency values in the figure with no decimals.
In any case, the frequencies in these spectra are used to give a first estimation of the hyperfine values which are then refined using HYSCORE results. The accuracy of the coupling parameters is determined by simulating the HYSCORE experiments.

• The orientation map is given with a + to – scale. There are no negative orientations
The + and - signs refer to more populated orientations and less populated orientations. We explain this in the current version of the manuscript.

• In panel b, the orientation map is cutting into the experimental EDNMR spectrum
The figure has been corrected so this does not happen in the new version.

HYSCORE
• Authors give no excitation bandwidth, in contrast, it is given for EDNMR after the revision
The excitation bandwith in HYSCORE experiments is 42 MHz since the excitation pulses are much shorter. This is now mentioned in the text.

Spin Hamiltonian:
• There is still an error in equation 1: I_j>1/2m nit I_i
The error has been corrected in the new version.

Further Comments:

• Figure 5 still talks about the CW spectrum.

The figure caption has been changed and now it reads "EPR absorption spectrum"

• The maximum of the EPR spectra significantly differs between the experiments and it is not clear why
The experiments were performed at slightly different frequencies.

REFEREE #2

The authors are greatly acknowledged for the effort they put into the review process to address the reviewers' comments.
We thank the reviewers for thoroughly revising our manuscript. We acknowledge the joint effort is benefitting the quality of the article.
The following further corrections are advised:

- Line 176: "Samples with a protein concentration of 400-800 mM". mM or µM? Please, check.
Thank you for realizing. There was a mistake in the "correction" in the first round. The actual concentration of the protein is 400-800 µM in a 50 mM MOPS buffer. Both mistakes have been corrected in the new version of the manuscript.

- Line 211: *db --> dB
This typo was corrected in the manuscript.

- Line 212: "The separation between the two pulses was tau = 1.5 µs". Between raising edges or between the falling edge of the HTA and the raising edge of the pi pulse? Please, clarify.
The time separation between the pulses is 1.5 µs between falling edge of the HTA and the raising edge of the pi pulse. This has been specified in the text.

- Line 232 (equation (1)): "sum over I(j)>1/2" rather "sum over I(i)>1/2".
Thank you for spotting this previously uncorrected mistake. It is now fixed.

- Line 242: "separate axis": unique axis?
We chose the wording *distinct axis* to make it more clear.

- Line 255: "C = h/(g_e·mu_B)" rather than "C = (g_e·mu_B)/h".
The referee is right, the mistake has been corrected.

- Line 312: *isoaloxacine --> isoalloxazine.
The spelling was corrected.

- Pages 15-16: Figure 5a appears to have been introduced twice.
That is true, it has been removed now. Thanks for noticing.

- Line 469: "Spectrum taken at the *tail of the EPR absorption" --> "Spectrum taken at the high-field tail of the EPR absorption".
Thanks for the suggestion, the sentence has been changed.

- Line 498: *isoallosazine --> isoalloxazine.
The spelling was corrected.

- Lines 542-545: the new sentence starting with "The overestimation in the 13C(4a) calculations" is not entirely clear to me. I would be grateful if the authors could consider reformulating it.
The sentence was changed and now reads: "The overestimation of the isotropic hyperfine coupling of $^{13}$C(4a) in the calculations is quite significant. While the

magnitude of the calculated isotropic couplings of $^{13}$C(4), $^{13}$C(5a) and $^{13}$C(10a) is comparable to the one of $^{13}$C(4a), the experimental hyperfine values reveal that the coupling with $^{13}$C(4a) is nearly half of the others."
We hope it is clearer now.

- Line 580: *isalloxacine --> isoalloxazine.
The spelling was corrected.

- Caption of Figure S4: the current text reads "HYSCORE simulation of [15N-FMN]-Fld variant at the high-field edge of the EPR spectrum". Please, check if the spectrum relates to the absorption maximum instead (see e.g. Figure 5b).
This was another mistake spotted by the referee. It is now corrected. Thanks!

---

## Author Response (AR1)

REPLY TO REVIEWERS

**Reviewer 1**

**Section "Experimental parameters":**

1.- Erratum in temperature (p. 362) has been corrected.

2.- HYSCORE experiments were measured at several tau values for the 13C-labelled samples (tau = 96, 112, 124, 176 and 144 ns for the central position, for example). In all of the spectra, the spectral features due to 13C(4) are very, very weak, yet, the spectrum shown in figure 4 of the manuscript (tau=112 ns) a bit better quality than the others. 13C features, which are very near the noise level show much better in the spectrum for tau=112 ns because it has a slightly better s/n ratio, that is why we decided to display it alone in the manuscript. The following figure, is the sum of the spectra corresponding to 124, 176 and 144 ns. As one can see, the 13C features are clearly there, but the spectrum in the manuscript is able to show them better. Moreover, no evident feature present in any other spectra is missing in the spectrum for tau 112 ns.

[Figure]

3.- To set and optimize the parameters, a single 1000 ns pulse was set and its power optimized to produce an FID. The FID integrated intensity was recorded as a function of the magnetic field. Then, the position of the magnetic field was fixed either at the center of the field-swept spectrum or at the high-end tail and an initial HTA ELDOR pulse was added. The ELDOR channel attenuation was initially set to 0 db, and several ELDOR-detected NMR spectra were taken varying the length of the ELDOR pulse from 1000 to 5000 ns. Then, the operation was repeated for several levels of ELDOR mw power. From the resulting spectra, the best s/n was found for HTA 1000 ns long and 0 db ELDOR power attenuation, so these parameters were adopted for longer accumulation of the spectra. The interpulse delay was chosen to be 1500 ns, long enough to let the potential FID of the first pulse decay. Information on the power of the HTA pulse and how the parameters were chosen has been added to the experimental section.

4.- Both, FID- detected and Echo-detected spectra were recorded. The referee is perfectly right, the FID-detected field-sweep experiments were recorded as a part of the set-up

procedure, but in the figure, echo-detected experiments are shown, just to indicate the field position where the ELDOR-detected NMR were taken.

**Section "Orientation Selección and Simulations:**

1.- Yes, the calculations of the orientation selection patterns is based on the simulation of the Q-band CW spectrum. An X-band simulation is now shown in Fig. 2 together with the experimental spectra.

2.- Misleading comment on orientation selection in Fig. 3 caption and text has been changed. The orientation selection patterns calculated for Q-band frequencies, and shown in the figure show the orientations being excited for the spectra recorded at the maximum of the EPR absorption and the high-field tail. The patterns are, again, based on the simulation of the spectrum at Q-band frequencies, so, they should be reliable. This pattern, for the magnetic field set to the maximum of the spectrum shows that all orientations are excited. While the orientations in the plane of the flavin ring contribute more, there is still some contribution of the orientations close to the perpendicular to the plane.

3.- The position of the nuclear features on the HYSCORE spectra were first calculated taking into account single nuclei one by one using *endorfreq*. Then, the simulation of the spectra was done with *saffron* using the complete set of nuclei which want to be simulated in order to obtain the feature's intensities and combination lines. We have done some rephrasing in the manuscript in order to clarify this paragraph, the referee was right in pointing out some confusion.

4.- In the HYSCORE simulations using saffron, orientation selection was indeed considered.

**Section "Experimental Hyperfina Spectra":**

- Experimental features described in the text has been highlighted (Figs. 3 and 4).

**Section "Error of measured parameters":**

 1.- Error in the expression (4) should be propagated from the individual errors to the subtraction. This is obtained from the squared root of the sum of the squared individual errors. The value $\pm 0.4$ mT is properly calculated.

2.- We have added on page 18 a brief explanation on how the errors in the parameters of the Spin Hamiltonian were estimated. The errors for the estimated quadrupole couplings have been added as well.

**Section "Discussion":**

- Following the reviewer's suggestion, discussion has been shortened and restructured in order to make it clearer.

**Section "Figure 1":**

- Suggested changes in the position and information collected in Fig. 1 have been implemented.

**Section "Spin Hamiltonian":**

- SH equations and subindexes for tensor principal values have been corrected, simplified and explained.

**Section "Further Comments":**

· Upon acceptance, raw data and simulation code will be made available in a open-access repository.

· The first sentence of the introduction has been split and the references cited next to the example.

· Typos have been corrected.

· The paragraph describing results and analysis of EDNMR experiments has been rewritten in order to make it more clear (see also "ELDOR-detected NMR "suggestions of reviewer 2).

· The description of the EDNMR experiment (former 260-264 lines) was rephrased.

· It is true that the intensities of the 13C(4) simulations do not reproduce well the experimental spectra. However the position of the HYSCORE features already allows estimating the magnitude of the hyperfine coupling of the nucleus with high precision.

**Reviewer 2**

**Specific comments**

Materials & Methods:

Point 1.- The applied phase cycle has been referred in the text.

Point 2.- Information on the power of the ELDOR pulse and the process of sequence optimization has now been included in the text.

Point 3.- The reference was added to the text. Fuchs et al. 2002.

ELDOR-detected NMR

Point1.- The text was changed in the manuscript to correct the mistake.

Point2.- The corrections have been implemented in the text. We thank the reviewer for spotting these errors.

Point3.- Although the study was performed at W-band, Davies ENDOR was used in Schleicher et al 2021 to obtain the couplings of 13C-labeled flavins but the signal of 13C(4) nucleus was not detected. The success of EDNMR as compared to ENDOR might have to do with the transitions being partially forbidden. This clarification has been added to the text.

Point 4.- The detection frequency was placed off center in the resonator dip. The measurements of both samples were tuned as similarly as possible in order to minimize problems with comparison of the spectra. We do not think the signal at 11 MHz is (at least entirely) an artifact of the subtraction spectrum due to different acquisition conditions, which were kept as close as possible, since in the parameters of 13C(4) obtained from HYSCORE spectra also predict one of the nuclear frequencies to be about this magnitude for orientations close to the parallel orientation. The width of the central hole in the spectrum is compatible with a 1 μs HTA pulse.

The referee is right about the labeling of the x-axis in the EDNMR spectra, we have changed it in the new version of the manuscript. We also have softened the statement about not having orientation selection at the center of the spectrum and we have detailed the spin Hamiltonian used for calculation of the orientation selection patterns. The patterns have been corrected since there was a mistake with the original excitation bandwidth, which is now explicitly mentioned.

Points 5, 6, 7, 8 and 10.- Description of the $^{13}$C(4a) features in the EDNMR spectrum and analysis has been revised and rewritten following the reviewer's suggestions in order to make it more accurate and more clear.

Point 9.- Several HTA pulse lengths and powers were tried, as indicated now I the Materials & Methods section. However these trial spectra were performed in order to spot the best s/n ratios in order to optimize the HTA pulse. Unfortunately, no detailed comparison was performed on the signal widths.

13C HYSCORE

Point 1.- Here is one full spectrum for [13C(2,4a)-FMN]-FLd

[Figure]

Far from being devoid of signals, the (-+) quadrant is dominated by 14N nitrogen signals but there is no evident 13C signals observed in the (-+) quadrant. A careful comparison between spectra of [13C(2,4a)-FMN]-FLd and [13C(2)-FMN]-FLd does not bring any signals that could be attributed to 13C(4a) in the (-+) quadrant. Therefore, when we wanted to focus of the 13C signals we have only shown only the (++) quadrant.

Point 3.- We agree with reviewer that representation in squared frequency axes would give a first estimation of the hyperfine parameters. We considered that we could skip this step of the analysis in the manuscript in order to make it lighter, as we already obtained a first estimation of the parameters form our X-band CW-EPR and Q-band EDNMR results that are enough for the simulation refinement.

Point 4.- Typo in line 334 has been corrected.

Point 5.- We have added on page 18 a brief explanation on how the uncertainties in the parameters of the Spin Hamiltonian were estimated.

Point 6.- Specific references to figures in the supplementary material have been added at all points were the Supplementary Material was referred to.

Points 7-8.- Reviewer's suggestions on Figs. S1 and S2 have been followed. Note that although the approximate field position with respect to the EPR spectrum is the same in Figs 3a & S2, the exact field position is different due to differences in the microwave frequencies. EDNMR and HYSCORE were performed at different spectrometers at different times.

$^{15}$N and $^{14}$N HSYCORE

Points 1 & 2.-Figures 5 and 6. Due to the reviewer's comment, all field positions at which the experiments were performed were checked and, indeed, some inconsistencies together with flat-out errors were found. For example, the magnetic field values of the EDNMR experiments were swap, so we realized the tail experiments were actually performed at the high-field tail and not the low-field tail of the EPR spectrum. After careful verification, we believe the field positions and corresponding diagrams are now correct in all figures. About the reason to choose the high-field end: Since the dominating anisotropy in the EPR spectrum is due to the axial hyperfine couplings of N(5), N(10), which are much larger in the direction perpendicular

to the isoaloxacine plane, positioning the magnetic field at any of those ends would select the "parallel" orientation. The high-field end was chosen because there is a small g-anisotropy, which at Qband gives a slightly better orientation selection for the high-field end.

Point 3.- In order to have the best possible orientation selection, we tried to do the experiments at the highest possible magnetic field that gives a good signal. For 14N signals this was possible at a quite high field. On the other hand, in the 15N labeled sample such a field position did not provide usable modulations and we had to move to a lower field. This is most probably related to the quadrupole coupling providing a way to mix the nuclear levels and yielding forbidden transitions that give good modulation of the echo. For 15N, the quadrupole interaction is missing and for an orientation very close to the perpendicular to the plane, which is an eigenaxis of the hyperfine coupling, the transitions are allowed and no modulation is observed.

The difference in the field setting between the spectra of the two samples is due to the difference in the microwave frequency of the two experiments.

Point 4.- The sum of the spectra was performed after Fourier transformation. Since the echo is strongly modulated, its intensity varies very much along the time trace. However, no significant difference in the envelope intensity is worth mentioning between tau = 96 ns and tau = 168 ns.

[Figure]

Point 5.- For the sake of clarity, in Figure 5 we have removed all antidiagonal lines except the one of $^{15}$N.

Point 6.- The sample is not deuterated but since some unidentified signals appear to lie on this diagonal, the antidiagonal line was drawn in the analysis phase. We have removed this antidiagonal line in order not to generate confusion.

Point 7.- After the display of the hyperfine parameters in the text, we have introduced a small paragraph explaining how the uncertainties were estimated. Within the mentioned uncertainty (+- 0.3 MHz) the shape of the correlation ridges is not compatible with rhombicity above the mentioned uncertainty. On the (++) quadrant, two separate and well defined peaks were found assigned to the parallel features of each of the two nitrogen nuclei. If there was any moderate rhombicity these features would be smeared out into a ridge.

We thank the reviewer's thorough comments, which allowed to identify and correct the above mentioned mistakes and increase the quality of the article.

Technical corrections.- All typos were corrected.

**Reviewer 3**

3rd paragraph (on additional hyperfine calculations).- We appreciate the reviewer's comments and agree with him that further effort should be made to improve the available calculations. On the other hand, we believe that such work is beyond the scope of the present manuscript. In this paper, we mainly show the experimental approach to obtain the hyperfine parameters for some specific nuclei in the flavoprotein. These show relevant differences with those predicted by the calculations, and this point is an important problem in two aspects: first, this probably indicates that the calculations do not adequately predict some details of the flavin structure within the protein; second, in some cases the calculated hyperfine parameters are used to understand specific properties, for example in the development of models for the magnetochemistry involved in the chemical compasses of birds, and these values could be inaccurate. Developing better calculations to overcome these problems is a very interesting open topic for future research, but we do not consider it for the present paper.

4th and 5th paragraphs (Statements and references on flavins in solution and semiquinone state in free flavins): Reference suggested by the reviewer were included and commented in the text.

6th paragraph (use of X-band CW-EPR spectra): We agree with the reviewer that the use of the CW-EPR spectrum to determine $A_z$ hyperfine parameter for $^{13}$C4a nucleus should require the support of simulations. However, in this case we only make use of this evidence to infer a first approximation of the parameter, wich is then refined from EDNMR and Q-band HYSCORE experiments. In the latter case, we do make use of the corresponding simulation to accurately determine the hyperfine parameters. On the other hand, most of the signal broadening of the X-band CW-EPR spectra is due to unresolved g anisotropy and dozens of weak hyperfine splittings that cannot be properly introduced into the simulation. Therefore, we consider that such simulations are not necessary in our case.

7th paragraph (Figs. 3 to 6): inserts related with orientation selection were changed to increase readibility.

On references: New references were added in all the places suggested by the reviewer.

Also typos and minor points were corrected.